# Impact of Evacuation on the Long-Term Trend of Metabolic Syndrome after the Great East Japan Earthquake

**DOI:** 10.3390/ijerph19159492

**Published:** 2022-08-02

**Authors:** Eri Eguchi, Narumi Funakubo, Hironori Nakano, Satoshi Tsuboi, Minako Kinuta, Hironori Imano, Hiroyasu Iso, Tetsuya Ohira

**Affiliations:** 1Department of Epidemiology, Fukushima Medical University School of Medicine, Fukushima 960-1295, Japan; naru-23@fmu.ac.jp (N.F.); h-nakano@fmu.ac.jp (H.N.); tsuboi@fmu.ac.jp (S.T.); teoohira@fmu.ac.jp (T.O.); 2Radiation Medical Science Center for the Fukushima Health Management Survey, Fukushima Medical University, Fukushima 960-1295, Japan; 3Department of Public Health, Okayama University Graduate School of Medicine, Dentistry and Pharmaceutical Sciences, Okayama 700-8558, Japan; kinuta@okayama-u.ac.jp; 4Public Health, Department of Social Medicine, Osaka University Graduate School of Medicine, Suita 565-0871, Japan; hiroimano@med.kindai.ac.jp (H.I.); hiso@it.ncgm.go.jp (H.I.); 5Department of Public Health, Kindai University Faculty of Medicine, Osakasayama 589-8511, Japan; 6Institute for Global Health Policy Research, Bureau of International Health Cooperation, National Center for Global Health and Medicine, Tokyo 162-8655, Japan

**Keywords:** disaster, Great East Japan Earthquake, disaster, metabolic syndrome, evacuation, Fukushima, national database

## Abstract

There has been an increase in lifestyle-related diseases in Fukushima Prefecture since the Great East Japan Earthquake. However, the overall long-term trends of lifestyle-related diseases in the Fukushima Prefecture according to the evacuation and other area are not reported. Therefore, we examined the long-term trends in the prevalence of metabolic syndrome before and after the Great East Japan Earthquake in Fukushima Prefecture according to these areas using a national database. The target population was approximately 330,000–440,000 per year; Fukushima Prefecture residents aged 40–74 years who underwent specific health check-ups during 2008–2017 participated in the study. Fukushima was divided into mountainous, central, coastal and evacuation areas. Using the Poisson regression model, the prevalence of metabolic syndrome in each fiscal year was determined by gender and age group for each location and compared before and after the disaster as well as between areas. Prevalence increased significantly throughout the observation period, particularly in the evacuation area. Age- and gender-adjusted prevalence rates significantly increased from 16.2% in 2010 to 19.5% in 2012 (prevalence ratios = 1.21) and 20.4% in 2017 in the evacuation area. Among other areas, coastal areas showed the highest increase with 17.9% (2017), followed by central areas with 16.5% (2017) and mountainous areas with 18.3% (2016). These increases were particularly high among men and the elderly. The prevalence of metabolic syndrome increased rapidly after the disaster, especially in evacuation area, and continued for subsequent 6–7 year. Long-term monitoring and measures to prevent lifestyle-related diseases are needed after major disasters, especially in evacuation areas, among men and the elderly.

## 1. Introduction

The Great East Japan Earthquake struck on 11 March 2011. In addition to the earthquake and tsunami, the nuclear accident caused a heavy burden on people in the Fukushima Prefecture, leading to the evacuation of 164,865 local residents (2012). According to studies conducted to date, major disasters in the United States and Japan have increased the prevalence of cardiovascular diseases (CVDs) and associated risk factors [1,2,3,4,5,6,7]. For example, in the high-impact area of Hurricane Sandy in New Jersey, it is reported that myocardial infarction incidence increased by 22%, 30-day mortality increased by 31% and stroke incidence increased by 7% compared to previous years [1]. In Japan, following the Hanshin-Awaji Earthquake, acute risk factors of CVDs during the period from night to morning, especially coronary heart disease deaths [2,3] and hypertension [4], increased for a few months after the earthquake. Furthermore, regarding the Great East Japan Earthquake in 2011, out-of-hospital cardiac arrest increased immediately after the earthquake in the Iwate, Miyagi and Fukushima prefectures approximately 1.7 times at 1 week after the earthquake to 1.26 times at 4 weeks after the earthquake [5]. Furthermore, the weekly occurrences of CVDs, including heart failure, acute coronary syndrome, stroke, cardiopulmonary arrest and pneumonia, in the Miyagi Prefecture were significantly increased after the Great East Japan Earthquake, compared with the past 3 years [6]; moreover, the number of patients with cardiovascular events in the Iwate Prefecture increased [7]. In Fukushima Prefecture, the prevalence of lifestyle-related disorders, such as CVDs and metabolic syndrome [8], hypertension [9], diabetes [10], dyslipidemia [11] and obesity [12], increased, particularly among the evacuees immediately after the disaster. For example, the prevalence of metabolic syndrome was 11.0% in men and 4.6% in women among non-evacuees in 2013, whereas it was 19.2% in men and 6.6% in women among evacuees [8], Regarding hypertension, the post-disaster changes in systolic and diastolic blood pressure among evacuees and nonevacuees were +5.8/3.4 and +4.6/2.1 mm Hg (*p* < 0.01/*p* < 0.0001) for men and +4.4/2.8 and +4.1/1.7 mm Hg (*p* = 0.33/*p* < 0.0001) for women, respectively [9]. Regarding diabetes, following the disaster, the prevalence significantly increased by approximately 1.8 points, and the authors observed that the incidence of diabetes was approximately 1.4 times greater among evacuees than among nonevacuees [10], Regarding dyslipidemia, the prevalence of hypo-high-density lipoprotein (HDL) cholesterolemia increased significantly from 6.0% to 7.2% following the disaster [11]. However, no study has examined the long-term trends of these cardiovascular risk factors, especially those related to metabolic syndrome, using a large-scale survey after the Great East Japan Earthquake. Although more than 10 years having passed since the disaster, more than 35,000 people in Fukushima have been forced to evacuate the region because of the nuclear accident (as of March 2022); hence, the possibility of prolonged and more pronounced increase in CVD risk factors as a result of the evacuation is a growing concern. This study used a national database (NDB) to analyze trend from 2008 to 2017 by each fiscal year (FY), according to evacuation area and other areas, in order to describe in detail the long-term trends in the prevalence of metabolic syndrome as a risk factor for CVD after the crisis. It is obvious that continuing support for the people of Fukushima and specific support for each background are required if metabolic syndrome, which was previously identified as a risk factor for lifestyle-related diseases in the immediate aftermath of the disaster, persists over the long term and its impact varies by evacuation status or other characteristics, The results of this study have important implications on measures to prevent potential future disasters, and they provide crucial insights into the long-term impact of evacuation due to disasters, especially earthquakes on metabolic syndromes and likely subsequent CVDs.

## 2. Materials and Methods

### 2.1. Study Population

The population aged 40–74 years in Fukushima Prefecture ranged from approximately 934,400 (2008) to 914,600 (2017). The number of participants in this study from Fukushima Prefecture per year consisted of approximately 340,000 (2008) to 440,000 (2017) people aged 40–74 years who took part in the specific health check-ups from 2008 to 2017 and whose data were recorded in the NDB. Of these, 3970 (2008) and 138 (2009–2017) were excluded because they lacked metabolic syndrome data, and approximately 333,400 (2008) to 439,700 (2017) participants with overlap were analyzed. The NDB contains all information on medical care covered by insurance, except for public assistance in Japan, and it stores all data on Japanese medical receipts and specific health check-ups. Specific health check-ups are medical examinations conducted in Japan since 2008, which are aimed at preventing and improving metabolic syndrome in all insured people and dependents aged 40–74 years.

### 2.2. Measurements

#### 2.2.1. Division of Fukushima Prefecture

Fukushima Prefecture has been historically divided into three regions (Figure 1) as follows: 1. the mountainous area (Aizu) is located in the most west side of the prefecture and the farthest from the nuclear power plant; 2. The central area (Nakadori) is located in the center of the prefecture and is a relatively larger city, and the capitals in prefectures, such as Koriyama City and Fukushima City, are in the area; 3. The coastal area (Hamadori) is the area along the easternmost coastline, including the nuclear power plant and most of the evacuation area. We divided the participants into four areas among these three areas in addition to the evacuation area based on registered postal code number. People from the evacuation area were excluded from other areas. The evacuation area included the following 12 places designated by the government for evacuation: 1. Tamura City, 2. Minamisoma City, 3. Kawamata-machi, 4. Hirono-machi, 5. Naraha-machi, 6. Tomioka-machi, 7. Kawauchi-mura, 8. Okuma-machi, 9. Futaba-machi, 10. Namie-machi, 11. Katsurao-mura, and 12. Iitate-mura.

#### 2.2.2. Classification of Metabolic Syndrome

Classification of metabolic syndrome according to the data of specific health check-ups from NDB were as follow: “1: Applicable to the criteria”, “2: Applicable to the pre-metabolic syndrome”, “3: Not applicable”, and “4: Undecidable”. We defined “1. Applicable to the criteria” as metabolic syndrome. The criteria of Japan Society for the Study of Obesity for metabolic syndrome were used for this classification, which are as follows: (1) abdominal circumference: ≥85 cm for men and ≥90 cm for women, (2) lipid: ≥150 mg/dL for neutral fat or <40 mg/dL for HDL cholesterol, (3) blood pressure: systolic ≥130 mmHg or diastolic ≥85 mmHg, and (4) fasting blood glucose level: ≥100 mg/dL or HbA1c > 5.6%. Among the participants, those who met the criteria for abdominal circumference and who met two or more of the other items (including each medication recipient) were categorized into “1: Applicable to the criteria”. For sensitivity analysis, the unified diagnostic criteria of six international academic societies were used as the criteria for metabolic syndrome, which were abdominal circumference: ≥90 cm for men and ≥80 cm for women, blood pressure: systolic of ≥130 mmHg and diastolic ≥85 mmHg, diabetes: fasting blood glucose level ≥ 100 mg/dL, neutral fat: ≥150 mg/dL, and HDL-C: <40 mg/dL for men and <50 mg/dL for women (including the users of each drug). Of these five items, those who fulfilled three or more items were categorized as having metabolic syndrome [13].

### 2.3. Statistical Analysis

First, we calculated the number of subjects, and participation rate according to each area and each FY. We also calculated the number of subjects and proportion of people aged ≥ 60 years according to gender. In this study, an age of 60 years, which is closer to the median, was considered as the cut-off point for the age category. Subsequently, we evaluated the prevalence of metabolic syndrome according to each area and each FY adjusted by gender and age categories (at every 5 years). The same calculation was conducted stratified by gender and age groups (≥60 and <60 years). Using Poisson regression analysis, the prevalence ratios (PRs) and 95% confidence intervals (CIs) of metabolic syndrome from 2011 to 2017 after the earthquake were calculated using the average prevalence of metabolic syndrome from 2008 to 2010 as a reference. The same analysis was conducted stratified by gender and age group (≥60 and <60 years). The PRs of metabolic syndrome (95% CIs) in the central, coastal, and evacuation areas, compared to those in the mountainous area, furthest from the nuclear power plant and least affected by the earthquake from before the earthquake in FY 2008 to after the earthquake in each FY till 2017 were also calculated. In addition, segmented log linear regression was used to determine and calculate the join point and annual percentage change. We conducted a one-tailed *t*-test for analyzing the difference in slope between sections to evaluate the trend change over the observation period. Moreover, as a sensitivity analysis, the same analysis was conducted using the unified diagnostic criteria of the six international academic societies as the criteria for metabolic syndrome. Covariates were gender and age categories (5 years each). SAS 9.4 and Joinpoint 4.7 (https://surveillance.cancer.gov/joinpoint/, accessed on 11, May 2022) were used for the analysis. The significance level of all analyses was set at α < 0.05. This study has been reviewed by the Ethics Committee of Fukushima Medical University (#30225).

## 3. Results

### 3.1. Characteristics of the Participants

Table 1 shows the characteristics of the study participants. The number of participants in Fukushima Prefecture was the highest in the central area (approximately 190,000–260,000), followed by the coastal area (approximately 50,000–80,000), mountainous area (approximately 50,000–66,000), and evacuation area (approximately 26,000–38,000). The proportions of people who participated in specific health check-ups in each area were 36.8–50.9% in the mountainous area, 37.3–49.3% in the central area, and 30.0–44.6% in the coastal area, comprising 36.3%, 39.0%, 38.5%, 27.9%, 33.5%, 35.9%, 37.9%, 39.1%, 39.8% and 41.1%, FYs 2008 to 2017, respectively, in the evacuation area. In the three areas other than the evacuation area, the participation rate slightly increased throughout the period. In the evacuation area, the proportion decreased by >10% in the year of the disaster in 2011 but returned to the original level in 2014 and then slightly increased to the level of other areas thereafter. The proportion of men was high in the evacuation area in the year of the disaster in 2011. The proportion of people aged ≥ 60 years was highest in the mountainous area before the disaster, but after 2012, the year after the disaster, the proportion became the highest in the evacuation area and then increased further.

### 3.2. Prevalence of Metabolic Syndrome

The trends in the proportion of metabolic syndrome are shown in Figure 2 and Figure 3. The proportion of metabolic syndrome showed an increasing trend in all four areas of the Fukushima prefecture since 2011—the year of the disaster, and the trend was especially significant in the evacuation area. The respective average annual percentage change from 2008–2017 were 1.1% (0.8–1.4%) for mountainous area, 0.8% (0.2–1.5%) for central area, 0.9% (0.5–1.3%) for coastal area and 2.9% (1.9–3.9%) for the evacuation area. In the evacuation area, the prevalence of metabolic syndrome was 16.2% in 2010, before the crisis, and ranged from 18.2% in 2008 to 20.4% in 2017 after the disaster. There was an inflection point in the central area in 2010 (i.e., the trend changed in 2010; *p* = 0.04), but there was no significant inflection point for other areas (proportions continuously increased throughout the period).

A similar trend was found in men and women, where the prevalence of metabolic syndrome showed increasing trends, especially in the evacuation area since 2011. The respective average annual percentage changes from 2008–2017 in the evacuation area were 3.1% (1.6–4.8%) for men and 3.3 (−0.1–6.8%) for women. The prevalence of metabolic syndrome were 22.8% in 2010 and 26.2% in 2008 to 29.3% in 2017 in men and 9.1% in 2010 and 9.6% in 2011, 11.8% in 2014, and 10.9% in 2017 in women. An inflection point was noted among men in 2012 (i.e., the trend changed in 2012; *p* = 0.03), but no significant inflection point was found among women. These trends were also similar in younger and older age groups, where the prevalence of metabolic syndrome showed increasing trends, especially in the evacuation area since 2011. The respective average annual percentage change from 2008–2017 in the evacuation area were 3.9% (2.3–5.5%) in the older age groups and 3.2% (1.1–5.3%) in the younger age groups. No inflection points were found among the younger and older age groups.

Compared to the prevalence before the disaster, the prevalence after the disaster was significantly high in the evacuation area. The PRs (95% CIs) in the period 2011–2017 compared to the mean prevalence in the period 2008–2010 was 1.13 (1.09–1.16) in 2011 and increased to 1.27 (1.23–1.30) in 2017. Similarly, this association was found in men and women, and in older and younger age groups (Table 2). Effect modification was found by gender on the association between the time and prevalence of metabolic syndrome in each area (*p* for interaction < 0.0001). Moreover, the effect modification of age groups on the association between the time and prevalence of metabolic syndrome in each area was significant in the central (*p* < 0.0001) and coastal areas (*p* < 0.01) but not significant in the mountainous (*p* = 0.79) and evacuation areas (*p* = 0.70).

The regional comparison are shown in Table 3. Compared with the mountainous area, the reference, the evacuation area showed significantly higher PRs of metabolic syndrome after the disaster, whereas no significant difference was found before the disaster. The PRs (95% CIs) were approximately 1.00 in 2008–2010, 1.08 in 2016 to 1.14 in 2012. These trends were also found in both men and women. The interactions between gender and area on the prevalence of metabolic syndrome were *p* = 0.02, <0.0001, and <0.0001 in 2008–2010 and 0.15, <0.0001, <0.01, 0.92, 0.94, 0.08, and <0.01 in 2011–2017, respectively. Compared with the mountainous area, the reference, the evacuation area also showed significantly higher PRs of metabolic syndrome after the disaster in both younger and older age groups, whereas no significant difference was found before the disaster. The modification effect of age groups on the association between the area and prevalence of metabolic syndrome was significant. The *p* values for interaction were <0.01, <0.01, <0.0001, <0.0001, <0.0001, <0.0001, <0.001, <0.01, <0.0001, <0.0001, and <0.0001, respectively, in 2008–2017, respectively.

The results of the sensitivity analysis conducted using international criteria for the evaluation of metabolic syndrome are shown in Appendix A. There were some minor differences in values, such as slightly larger PR differences, between men and women in the comparison of values before and after the disaster, but the overall trend was similar to the original result.

## 4. Discussion

We investigated the trends in the prevalence of metabolic syndrome before and after the disaster in four regions of the Fukushima Prefecture according to gender and age group. The prevalence of metabolic syndrome in the Fukushima Prefecture was found to be significantly higher after the disaster than before the disaster, and this trend was particularly prominent in the evacuation area and it did not decrease until 2017. Furthermore, the prevalence of metabolic syndrome was significantly higher in the evacuation area than in the other areas in the prefecture after the disaster.

In the United States, it has been reported that the incidence of CVD and its risk factors increased immediately after a large-scale disaster [1]. In Japan, an increase in the incidence of CVD and its risk factors have been reported after the Great Hanshin-Awaji Earthquake [2,3,4]. Studies have also reported that the incidence of cardiovascular disease and its risk factors increased immediately after the Great East Japan Earthquake [5,6,7]. In contrast, regarding for long-term health outcomes following the disaster, authors of the systematic review on 58 articles reviewing post-disaster health hazards concluded that attention is needed to the negative indirect long-term effects on cardiometabolic health; the majority of these articles reported increased CVD, mortality, and associated risk factors, such as diabetes and obesity [14]. For example, a 12-year follow-up of 21,079 pre-disaster and 84,751 post-disaster individuals following the 2005 Katrina disaster reported an increase in post-disaster acute myocardial infarction, or cardiac death, diabetes and dyslipidemia [15] and emphasized the importance of long-term monitoring of post-disaster health. However, there was no long-term follow-up study of lifestyle-related disease trends after the 2011 Great East Japan Earthquake, tsunami and nuclear accident, that compared evacuated areas to other areas.

To the best of our knowledge, because we were able to show for the first time that the prevalence of metabolic syndrome increased following a major disaster but did not decline for a specific amount of time despite long-term surveillance, the findings of this study are novel.

The present study did not clearly elucidate the mechanism of the trends by which the incidence of lifestyle-related diseases increased after the disaster and has not declined since then. However, the first possible explanation is the change in lifestyle due to the disaster. According to the Fukushima Health Management Survey, approximately 30% of evacuees lost their jobs after the disaster, which may have affected their health through lack of physical activity and other lifestyle changes. Reports from the same survey showed that in the immediate aftermath of the disaster, in FY2011 and FY2012, there was an increase in the incidence of metabolic syndrome [8], hypertension [9], diabetes [10], dyslipidemia [11] and obesity [12]. It is conceivable that these lifestyle-related diseases, which showed an increased incidence due to lifestyle changes, may not have improved since then. In fact, it has been reported that the amount of physical activity was reduced in temporary housing residents [16]. Moreover, a study reported that juice-based dietary patterns that changed after the disaster were associated with the subsequent development of dyslipidemia and other diseases [17]. These changes in lifestyle are likely have an impact. The second mechanism is the deterioration of mental health. It has been reported that natural disasters, such as earthquakes and extreme stress, in other countries trigger acute myocardial infarction and sudden cardiac death [18]. After earthquakes in Croatia, increased rates of suicidal tendency, susceptibility to post traumatic stress disorder, and depressive symptoms were reported followed by an increased numbers of patients with gastric and duodenal ulcers, abdominal pain, bloating, constipation, and acute myocardial infarction due to cardiovascular events caused by acute stress, stroke, arrhythmia, and tachycardia. Increased incidence of cardiomyopathy and other disorders have been reported, indicating the importance of treating psychological and psychosomatic effects [19]. The Fukushima Health Management Survey also reported that 14.6% of the participants in 2012 experienced psychological distress [20], which deteriorated their eating habits [21,22], smoking [23], drinking behavior [24] and sleep disturbance [25]. Furthermore, those who changed their socioeconomic activities were reported to have lower subjective health and trauma-related reactions [26]. There exists a possibility that metabolic syndrome was affected by these psychological factors. Ten years have elapsed since the occurrence of the Great East Japan Earthquake, but in Fukushima prefecture, the direct damage from the radiation disaster still continues, with more than 35,000 people still evacuating (as of March 2022). Studies conducted to date have reported that the abovementioned effects on lifestyle and mental health are more serious among evacuees. For instance, a higher rate of psychological distress [27] and a lower rate of exercise habits were reported in evacuees [28,29]. These reports are consistent with the results of the present study, with a higher prevalence of metabolic syndrome in the evacuation area. In addition, various environmental and anthropogenic characteristics associated with evacuation may be interrelated and influence the increased prevalence of metabolic syndrome.

The results of this study suggest that the long-term impact of disasters on metabolic syndrome was greater in males than in females and, in some areas, the impact was greater in older than in younger age groups. The above impacts on work, lifestyle, and mental health, and health attitudes may differ across gender and age groups and should be used in future lifestyle-related disease disaster prevention responses by gender and age groups.

The comparison of results of this study with the results of the previous Fukushima health management survey and other reports demonstrates that the various effects of the disaster affected lifestyle-related diseases through lifestyle changes and mental health deterioration, which did not improve. Moreover, even if they improved, they did not result in an improvement of these diseases, which could be the cause of the prevalence of these diseases.

There is a concern that the incidence of lifestyle-related diseases, such as CVDs [30], dementia, and so on will continue to be higher among people in evacuation areas, and that support for people in evacuation areas, men, and elderly, should be strengthened. In addition, it is important to continuously monitor and explore the detailed risk factors in people in Fukushima for the prevention of these diseases. Moreover, it is clear that long-term support, especially for people in the evacuation area, men, and elderly, is needed in the event of a disaster in the future.

The strengths of this study are threefold: first, it is large-scale research conducted using data collected from residents in Fukushima who underwent specific health check-ups. Second, we could stratify and analyze the data for the evacuation area and the three regions in Fukushima Prefecture. Third, the existence of large-scale data before and after the disaster enabled making comparisons before and after the disaster, which was a considerable advantage. However, one of the limitations of this study is that we did not have the information regarding whether people have actually evacuated because the classification of the area was conducted on the basis of the participants’ address. Of the 12 cities, towns, and villages in the evacuation area in this study, all residents of 9 cities, towns, and villages have evacuated. In three cities and towns, only some areas have received evacuation orders and, therefore, they included people who have not evacuated. Hence, it is possible that the actual impact of evacuation is underestimated.

## 5. Conclusions

In the Fukushima prefecture, the prevalence of metabolic syndrome was significantly higher after the disaster than before the disaster in both men and women as well as young and old age groups, and it has not improved. This trend was particularly predominant in the evacuation area. Currently, 10 years after the disaster, it is important to continue to monitor the health status of people in Fukushima, especially in the evacuation area. We should further explore the risk factors for metabolic syndrome in relation to other lifestyle-related diseases in detail.

## Figures and Tables

**Figure 1 ijerph-19-09492-f001:**
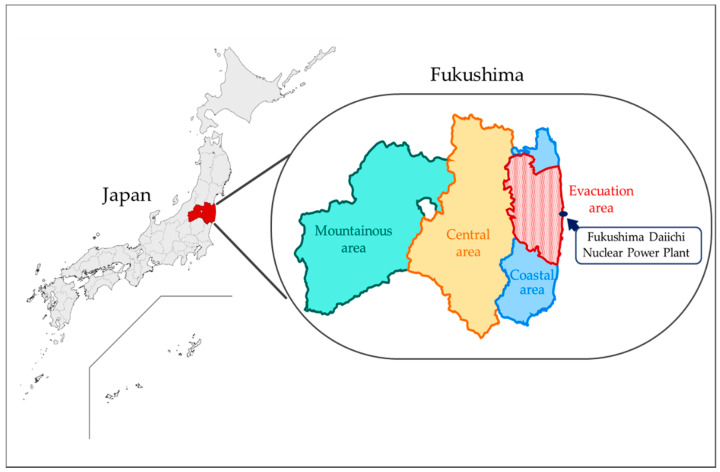
Mountainous area, central area, coastal area and evacuation area in Fukushima prefecture. Map of Japan highlighting Fukushima Prefecture. This study included all areas in Fukushima: Mountainous area (Aizu), central area (Naka-dori), coastal area (Hama-dori), and evacuation area.

**Figure 2 ijerph-19-09492-f002:**
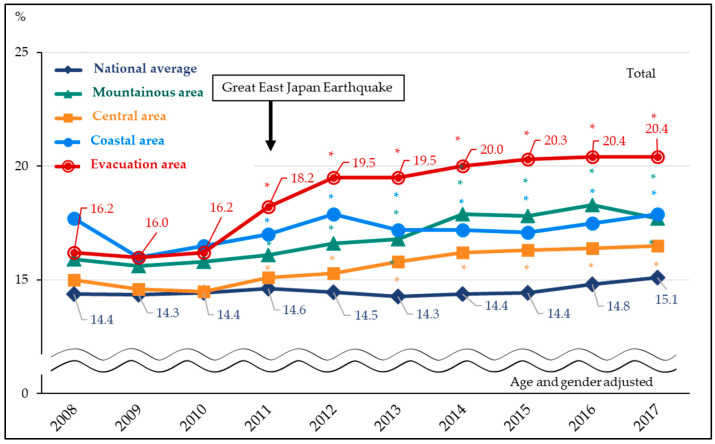
Total prevalence of metabolic syndrome before and after the disaster (fiscal year 2008–2017). Trends in metabolic syndrome prevalence before (2008–2010) and after (2011–2017) the earthquake among four Fukushima areas and the national average. Significant differences in metabolic syndrome prevalence ratios compared with the reference period (2008–2010) are indicated using an asterisk (*). All probability values for statistical tests were two-tailed, and *p* < 0.05 was regarded as statistically significant.

**Figure 3 ijerph-19-09492-f003:**
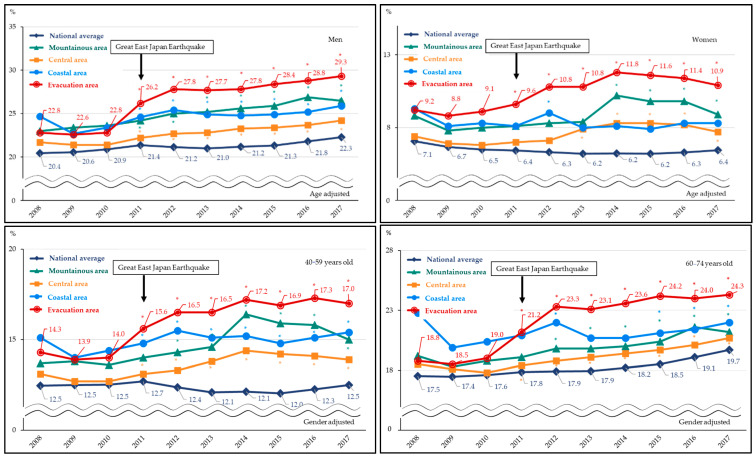
Prevalence of metabolic syndrome before and after the disaster according to gender and age categories (fiscal year 2008–2017). Trends in metabolic syndrome prevalence before (2008–2010) and after (2011–2017) the earthquake among four Fukushima areas and the national average in men and women, and in older and younger age groups. Significant differences in metabolic syndrome prevalence ratios compared with the reference period (2008–2010) are indicated using an asterisk (*). All probability values for statistical tests were two-tailed, and *p* < 0.05 was considered as statistically significant.

**Table 1 ijerph-19-09492-t001:** Numbers and age prevalence of participants’ characteristics in each fiscal year.

	2008	2009	2010	2011	2012	2013	2014	2015	2016	2017
**Total**										
**Numbers**	333,225	357,731	359,649	357,276	376,115	386,279	407,489	417,071	428,048	439,683
**Mountainous area**	51,658	53,220	55,258	58,164	59,699	59,931	63,964	64,354	64,952	65,819
**Central area**	191,088	206,178	204,343	213,036	220,794	225,718	237,751	242,193	249,240	256,728
**Coastal area**	54,978	60,542	63,072	59,562	64,619	67,687	71,061	74,716	77,537	79,990
**Evacuation area**	35,501	37,791	36,976	26,514	31,003	32,943	34,713	35,808	36,319	37,146
**Participation rate, %**										
**Mountainous area**	36.8	38.4	40.4	43.0	44.6	45.2	48.4	49.0	49.9	50.9
**Central area**	37.3	40.2	39.9	41.7	43.3	44.1	46.2	46.7	47.9	49.3
**Coastal area**	30.0	33.2	34.7	33.0	36.1	37.8	39.5	41.5	43.1	44.6
**Evacuation area**	36.3	39.0	38.5	27.9	33.5	35.9	37.9	39.1	39.8	41.1
**Men**										
**Numbers**										
**Mountainous area**	24,871	25,855	27,062	28,830	29,798	29,998	32,413	32,872	33,093	33,612
**Central area**	101,621	108,877	107,738	112,592	117,051	119,789	126,927	128,099	131,789	135,088
**Coastal area**	29,115	32,624	34,131	32,605	35,315	37,041	38,930	40,738	42,132	43,518
**Evacuation area**	17,818	18,999	18,593	14,187	16,212	17,211	17,936	18,423	18,815	19,288
**Age ≥ 60, %**										
**Mountainous area**	36.6	37.7	37.4	37.5	38.2	39.4	39.0	39.3	39.7	40.1
**Central area**	29.8	30.3	32.0	32.0	32.9	33.5	33.9	34.7	34.9	35.1
**Coastal area**	30.5	29.7	30.9	30.9	31.2	32.4	33.6	33.6	33.8	34.2
**Evacuation area**	37.0	38.0	38.6	36.4	41.5	42.4	43.6	44.4	45.1	45.3
**Women**										
**Numbers**										
**Mountainous area**	26,787	27,365	28,196	29,334	29,901	29,933	31,551	31,482	31,859	32,207
**Central area**	89,467	97,301	96,605	100,444	103,743	105,929	110,824	114,094	117,451	121,640
**Coastal area**	25,863	27,918	28,941	26,957	29,304	30,646	32,131	33,978	35,405	36,472
**Evacuation area**	17,683	18,792	18,383	12,327	14,791	15,732	16,777	17,385	17,504	17,858
**Age ≥ 60, %**										
**Mountainous area**	46.4	46.0	46.1	45.8	45.7	46.5	45.6	45.9	45.6	45.5
**Central area**	40.3	39.4	40.4	40.4	40.7	41.4	41.5	42.2	42.0	41.9
**Coastal area**	42.5	41.3	42.3	41.2	41.0	42.5	42.8	42.4	41.5	41.5
**Evacuation area**	44.5	43.3	44.3	43.7	48.4	50.2	50.3	49.9	51.6	51.9

**Table 2 ijerph-19-09492-t002:** Risk ratios and 95% confidence intervals of metabolic syndrome after the Great East Japan Earthquake compared to that before the disaster.

	2008–2010	2011	2012	2013	2014	2015	2016	2017
**Total**								
**Mountainous area**	1.00	1.02	1.05	1.06	1.13	1.13	1.16	1.12
		(1.00–1.05)	(1.03–1.08)	(1.04–1.08)	(1.11–1.16)	(1.10–1.15)	(1.14–1.19)	(1.10–1.15)
** Central area**	1.00	1.02	1.04	1.07	1.10	1.10	1.11	1.12
		(1.01–1.03)	(1.02–1.05)	(1.05–1.08)	(1.09–1.11)	(1.09–1.12)	(1.10–1.12)	(1.10–1.13)
** Coastal area**	1.00	1.02	1.07	1.02	1.02	1.02	1.04	1.06
		(0.99–1.04)	(1.04–1.09)	(1.00–1.04)	(1.00–1.05)	(1.00–1.04)	(1.02–1.06)	(1.04–1.09)
** Evacuation area**	1.00	1.13	1.21	1.21	1.24	1.26	1.27	1.27
		(1.09–1.16)	(1.18–1.25)	(1.18–1.25)	(1.21–1.28)	(1.22–1.29)	(1.23–1.30)	(1.23–1.30)
**Men**								
** Mountainous area**	1.00	1.04	1.07	1.08	1.10	1.11	1.15	1.13
		(1.01–1.07)	(1.04–1.10)	(1.05–1.11)	(1.07–1.13)	(1.08–1.14)	(1.12–1.18)	(1.10–1.16)
** Central area**	1.00	1.03	1.06	1.06	1.08	1.08	1.10	1.12
		(1.02–1.05)	(1.04–1.07)	(1.04–1.07)	(1.07–1.09)	(1.07–1.10)	(1.08–1.11)	(1.11–1.14)
** Coastal area**	1.00	1.05	1.08	1.05	1.05	1.05	1.07	1.09
		(1.02–1.07)	(1.05–1.11)	(1.03–1.08)	(1.03–1.08)	(1.03–1.08)	(1.04–1.09)	(1.07–1.12)
** Evacuation area**	1.00	1.15	1.22	1.22	1.22	1.24	1.26	1.28
		(1.11–1.20)	(1.18–1.27)	(1.18–1.26)	(1.18–1.26)	(1.21–1.29)	(1.22–1.30)	(1.24–1.32)
**Women**								
** Mountainous area**	1.00	0.98	1.01	1.03	1.25	1.21	1.20	1.10
		(0.94–1.03)	(0.97–1.06)	(0.98–1.07)	(1.20–1.31)	(1.16–1.26)	(1.15–1.25)	(1.05–1.14)
** Central area**	1.00	1.00	1.00	1.12	1.18	1.18	1.16	1.10
		(0.97–1.03)	(0.97–1.03)	(1.09–1.14)	(1.15–1.21)	(1.15–1.21)	(1.13–1.19)	(1.07–1.12)
** Coastal area**	1.00	0.94	1.05	0.94	0.95	0.93	0.97	0.97
		(0.90–0.99)	(1.00–1.10)	(0.90–0.98)	(0.91–0.99)	(0.89–0.97)	(0.93–1.01)	(0.93–1.02)
** Evacuation area**	1.00	1.06	1.20	1.20	1.31	1.29	1.27	1.21
		(0.99–1.13)	(1.13–1.27)	(1.14–1.27)	(1.24–1.38)	(1.22–1.36)	(1.21–1.34)	(1.15–1.28)
**Age < 60 years**								
** Mountainous area**	1.00	1.02	1.04	1.06	1.19	1.16	1.15	1.09
		(0.99–1.06)	(1.01–1.08)	(1.03–1.10)	(1.16–1.23)	(1.12–1.20)	(1.12–1.19)	(1.06–1.13)
** Central area**	1.00	1.02	1.03	1.07	1.12	1.10	1.10	1.08
		(1.00–1.04)	(1.02–1.05)	(1.06–1.09)	(1.10–1.14)	(1.09–1.12)	(1.08–1.12)	(1.06–1.10)
** Coastal area**	1.00	1.03	1.07	1.05	1.05	1.02	1.04	1.06
		(1.00–1.06)	(1.04–1.10)	(1.02–1.08)	(1.02–1.08)	(0.99–1.05)	(1.02–1.07)	(1.03–1.09)
** Evacuation area**	1.00	1.11	1.18	1.18	1.22	1.21	1.23	1.21
		(1.06–1.16)	(1.13–1.23)	(1.13–1.23)	(1.17–1.27)	(1.16–1.26)	(1.18–1.28)	(1.17–1.26)
**Age ≥ 60 years**								
** Mountainous area**	1.00	1.02	1.06	1.05	1.07	1.09	1.15	1.13
		(0.98–1.05)	(1.02–1.09)	(1.02–1.09)	(1.03–1.10)	(1.05–1.12)	(1.12–1.19)	(1.10–1.16)
** Central area**	1.00	1.02	1.04	1.05	1.07	1.09	1.11	1.14
		(1.00–1.04)	(1.02–1.06)	(1.03–1.07)	(1.05–1.09)	(1.07–1.11)	(1.09–1.13)	(1.12–1.16)
** Coastal area**	1.00	1.00	1.05	0.98	0.99	1.00	1.02	1.05
		(0.96–1.03)	(1.01–1.08)	(0.95–1.02)	(0.96–1.02)	(0.97–1.04)	(0.99–1.05)	(1.02–1.08)
** Evacuation area**	1.00	1.13	1.24	1.23	1.26	1.29	1.28	1.29
		(1.08–1.19)	(1.19–1.29)	(1.19–1.28)	(1.21–1.30)	(1.24–1.34)	(1.23–1.33)	(1.25–1.34)

**Table 3 ijerph-19-09492-t003:** Risk ratios of metabolic syndrome in the evacuation area compared with the mountainous area in fiscal year 2008–2017.

	2008	2009	2010	2011	2012
**Total**					
** Mountainous area**	1.00	1.00	1.00	1.00	1.00
** Central area**	0.95 (0.92–0.97)	0.94 (0.91–0.96)	0.92 (0.90–0.95)	0.93 (0.91–0.96)	0.92 (0.90–0.94)
** Coastal area**	1.11 (1.08–1.14)	1.02 (0.99–1.05)	1.04 (1.01–1.07)	1.05 (1.02–1.08)	1.07 (1.04–1.10)
** Evacuation area**	0.99 (0.96–1.03)	0.99 (0.96–1.02)	1.00 (0.97–1.03)	1.09 (1.06–1.13)	1.14 (1.11–1.18)
**Men**					
** Mountainous area**	1.00	1.00	1.00	1.00	1.00
** Central area**	0.97 (0.94–0.99)	0.94 (0.91–0.96)	0.93 (0.90–0.96)	0.94 (0.91–0.96)	0.93 (0.91–0.96)
** Coastal area**	1.11 (1.07–1.14)	0.99 (0.96–1.03)	1.02 (0.99–1.05)	1.04 (1.01–1.08)	1.05 (1.02–1.08)
** Evacuation area**	0.98 (0.94–1.02)	0.96 (0.92–1.00)	0.96 (0.92–1.00)	1.07 (1.03–1.12)	1.10 (1.06–1.14)
**Women**					
** Mountainous area**	1.00	1.00	1.00	1.00	1.00
** Central area**	0.91 (0.87–0.95)	0.95 (0.90–0.99)	0.92 (0.87–0.96)	0.93 (0.89–0.97)	0.91 (0.87–0.95)
** Coastal area**	1.12 (1.06–1.18)	1.10 (1.04–1.17)	1.10 (1.04–1.16)	1.05 (0.99–1.12)	1.14 (1.08–1.21)
** Evacuation area**	1.04 (0.98–1.11)	1.11 (1.05–1.19)	1.13 (1.06–1.21)	1.18 (1.10–1.26)	1.29 (1.21–1.37)
**Age < 60 years**					
** Mountainous area**	1.00	1.00	1.00	1.00	1.00
** Central area**	0.92 (0.89–0.96)	0.89 (0.86–0.92)	0.90 (0.87–0.93)	0.90 (0.87–0.93)	0.90 (0.87–0.93)
** Coastal area**	1.04 (1.00–1.08)	0.95 (0.91–0.99)	0.99 (0.96–1.03)	1.00 (0.96–1.04)	1.02 (0.98–1.06)
** Evacuation area**	1.03 (0.98–1.08)	0.99 (0.95–1.04)	1.01 (0.96–1.06)	1.09 (1.04–1.15)	1.13 (1.08–1.19)
**Age ≥ 60 years**					
** Mountainous area**	1.00	1.00	1.00	1.00	1.00
** Central area**	0.95 (0.92–0.99)	0.97 (0.94–1.01)	0.93 (0.90–0.96)	0.95 (0.92–0.98)	0.93 (0.90–0.96)
** Coastal area**	1.17 (1.12–1.22)	1.07 (1.02–1.12)	1.06 (1.02–1.11)	1.07 (1.03–1.12)	1.09 (1.05–1.13)
** Evacuation area**	0.97 (0.92–1.01)	1.00 (0.95–1.05)	0.99 (0.95–1.04)	1.09 (1.04–1.15)	1.15 (1.10–1.20)
	**2013**	**2014**	**2015**	**2016**	**2017**
**Total**					
** Mountainous area**	1.00	1.00	1.00	1.00	1.00
** Central area**	0.94 (0.92–0.96)	0.91 (0.89–0.93)	0.91 (0.90–0.93)	0.90 (0.88–0.91)	0.93 (0.91–0.95)
** Coastal area**	1.01 (0.99–1.04)	0.95 (0.93–0.98)	0.95 (0.93–0.98)	0.94 (0.92–0.97)	1.00 (0.98–1.03)
** Evacuation area**	1.13 (1.10–1.17)	1.09 (1.06–1.12)	1.11 (1.07–1.14)	1.08 (1.05–1.11)	1.12 (1.09–1.15)
**Men**					
** Mountainous area**	1.00	1.00	1.00	1.00	1.00
** Central area**	0.93 (0.91–0.95)	0.93 (0.91–0.95)	0.93 (0.91–0.95)	0.90 (0.88–0.93)	0.94 (0.92–0.96)
** Coastal area**	1.02 (0.99–1.05)	1.00 (0.97–1.03)	0.99 (0.96–1.02)	0.97 (0.94–0.99)	1.01 (0.98–1.04)
** Evacuation area**	1.09 (1.05–1.13)	1.07 (1.03–1.11)	1.08 (1.05–1.12)	1.05 (1.02–1.09)	1.09 (1.05–1.12)
**Women**					
** Mountainous area**	1.00	1.00	1.00	1.00	1.00
** Central area**	0.99 (0.94–1.03)	0.85 (0.81–0.88)	0.88 (0.85–0.92)	0.88 (0.84–0.91)	0.91 (0.87–0.95)
** Coastal area**	1.00 (0.95–1.06)	0.82 (0.78–0.87)	0.83 (0.79–0.88)	0.88 (0.84–0.93)	0.97 (0.92–1.02)
** Evacuation area**	1.27 (1.20–1.35)	1.14 (1.08–1.20)	1.17 (1.10–1.23)	1.16 (1.09–1.22)	1.20 (1.13–1.27)
**Age < 60 years**					
** Mountainous area**	1.00	1.00	1.00	1.00	1.00
** Central area**	0.92 (0.89–0.95)	0.85 (0.83–0.88)	0.87 (0.84–0.89)	0.87 (0.84–0.89)	0.89 (0.87–0.92)
** Coastal area**	0.98 (0.95–1.02)	0.88 (0.85–0.91)	0.88 (0.85–0.91)	0.91 (0.87–0.94)	0.97 (0.93–1.00)
** Evacuation area**	1.12 (1.07–1.17)	1.03 (0.99–1.08)	1.05 (1.01–1.10)	1.08 (1.03–1.13)	1.12 (1.07–1.17)
**Age ≥ 60 years**					
** Mountainous area**	1.00	1.00	1.00	1.00	1.00
** Central area**	0.95 (0.92–0.98)	0.95 (0.93–0.98)	0.95 (0.93–0.98)	0.92 (0.89–0.94)	0.96 (0.93–0.99)
** Coastal area**	1.02 (0.99–1.06)	1.02 (0.98–1.06)	1.02 (0.98–1.05)	0.97 (0.94–1.01)	1.02 (0.99–1.06)
**Evacuation area**	1.15 (1.10–1.20)	1.15 (1.11–1.20)	1.16 (1.12–1.21)	1.09 (1.05–1.13)	1.12 (1.08–1.17)

## Data Availability

The data used in this study were obtained from the Japan Ministry of Health, Labour and Welfare through formal procedures. (https://www.mhlw.go.jp/stf/seisakunitsuite/bunya/kenkou_iryou/iryouhoken/reseputo/index.html accessed on 26 July 2022).

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
