# Peer review of "Impact of Evacuation on the Long-Term Trend of Metabolic Syndrome after the Great East Japan Earthquake"

_ijerph, 2022, doi:10.3390/ijerph19159492_

Round 1
Reviewer 1 Report
Thank you for giving me the opportunity to review this valuable manuscript. This research gives an insight on how natural disasters pose a threat to cardiovascular risk factors.I have a few comments/suggestions/ questions for the authors
1- In the introduction section, the authors briefly state that the lifestyle-related disorders has increased after the disaster. Could please elaborate on this? (e.g., provide numbers)
2- in the methods sections, please delete the sentences that have population data across the years (e.g., 934,393, 930,254, 926,424, 922,040, 914,941, 911,801, 914,999, 918,058, 917,182, and 914,613).
These numbers are confusing, and unnecessary in text. You can refer to Table 1 for these numbers.
Similarly, the PRs values in the results sections are unnecessary in text (e.g., The PRs (95% CIs) were 0.99 (0.96–1.03), 0.99 (0.96–1.02), and 1.00 (0.97–1.03) in 2008–2010, respectively, and 1.09 (1.06–1.13), 1.14 (1.11–1.18), 1.13 (1.10–1.17), 1.09 (1.06–1.12), 1.11 (1.07–1.14), 1.08 (1.05–1.11), and 1.12 (1.09–1.15). you can refer to Table 3 instead.
3- in the statistical analysis section
"First, we calculated the number of subjects, participation rate, prevalence of men, and prevalence of people aged ≥60 years"
Can you please explain to the readers why did you chose the age of 60 years as a cut point. I do not think it was mentioned anywhere in text?
4- Table 1 is very confusing, I highly recommend revising it for clarity to the readers.
5- Table 3: I do not understand why risk ratios of metabolic syndrome in the evacuation area was compared with the Aizu area in particular.
6- I think the results of this paper are valuable, however, the paper is missing a clear implications and recommendations to reduce cardiovascular risk factors following a disaster.
Author Response
Response to Comments
We would like to thank you sincerely for your valuable and useful comments. We have taken your comments into account and have revised the manuscript based on your advice. We hope that you will read our response and recheck the manuscript.
Reviewer #1
Comment #1
- In the introduction section, the authors briefly state that the lifestyle-related disorders has increased after the disaster. Could please elaborate on this? (e.g., provide numbers)
Response #1
Thank you for pointing that out. As you pointed out, we have added more detail in the background section regarding the increase in lifestyle-related diseases after the disaster. They are as follows. “Studies conducted to date have reported that the incidence of cardiovascular disease and its risk factors has increased after a large-scale disaster in the United states and Japan [1–7]. For example, in the high-impact area of Hurricane Sandy in New Jersey, It is reported that myocardial infarction incidence increased by 22% compared to previous years, with a 31% increase in 30-day mortality, the incidence of stroke increased by 7% [1]. In Japan, following the Hanshin-Awaji Earthquake, it was found that acute risk factors of cardiovascular disease during the period from night-time to morning, especially Coronary heart disease deaths [2, 3] and hypertension [4] increased for a few months after the earthquake. Furthermore, regarding the Great East Japan Earthquake in 2011, it was reported that out-of-hospital cardiac arrest increased immediately after the earthquake in Iwate, Miyagi and Fukushima prefectures approximately 1.7 times (1 week after the earthquake) to 1.26 times (4 weeks after the earthquake) [5] Furthermore, The weekly occurrences of CVDs, including heart failure, acute coronary syndrome, stroke, cardiopulmonary arrest, and pneumonia n Miyagi Prefecture were all significantly increased after the Great East Japan Earthquake compared with the previous 3 years [6] and that the number of patients with cardiovascular events in Iwate Prefecture was also increased [7]. In Fukushima prefecture, it has been reported that the proportion of lifestyle-related disorders such as cardiovascular diseases and metabolic syndrome [8], hypertension [9], diabetes [10], dyslipidemia [11], and obesity [12] has increased especially in the evacuees immediately after the disaster. For example, the prevalence of metabolic syndrome was 11.0% in men and 4.6% in women among non-evacuee in 2013, while they were 19.2% in men and 6.6% in women among evacuees [8], and for hypertension, the post-disaster changes in systolic and diastolic blood pressure among the evacuees and non-evacuees were +5.8/3.4 versus +4.6/2.1 mm Hg (P<0.01/P<0.0001) for men and +4.4/2.8 versus +4.1/1.7 mm Hg (P=0.33/P<0.0001) for women, respectively [9]. In diabetes, after the disaster, the prevalence significantly increased around 1.8 points, and authors observed that the incidence of diabetes was around 1.4 times greater among evacuees than among non-evacuees [10], and in lipid abnormalities, it has been reported that the prevalence of hypo-high-density lipoprotein (HDL) cholesterolemia increased significantly from 6.0% to 7.2% following the disaster [11].” (P2 L48-77)
Comment #2
- in the methods sections, please delete the sentences that have population data across the years (e.g., 934,393, 930,254, 926,424, 922,040, 914,941, 911,801, 914,999, 918,058, 917,182, and 914,613).
These numbers are confusing, and unnecessary in text. You can refer to Table 1 for these numbers.
Similarly, the PRs values in the results sections are unnecessary in text (e.g., The PRs (95% CIs) were 0.99 (0.96–1.03), 0.99 (0.96–1.02), and 1.00 (0.97–1.03) in 2008–2010, respectively, and 1.09 (1.06–1.13), 1.14 (1.11–1.18), 1.13 (1.10–1.17), 1.09 (1.06–1.12), 1.11 (1.07–1.14), 1.08 (1.05–1.11), and 1.12 (1.09–1.15). you can refer to Table 3 instead.
Response #2
Thank you for pointing that out. We have corrected it accordingly. (P5 L179-180, P10 L250-253).
Comment #3
- in the statistical analysis section
"First, we calculated the number of subjects, participation rate, prevalence of men, and prevalence of people aged ≥60 years"
Can you please explain to the readers why did you chose the age of 60 years as a cut point. I do not think it was mentioned anywhere in text?
Response #3
Thank you for your important point, we have added the following explanation as to why we have set the cut-off at age 60 “In the present study, an age of 60 years, which is closer to the median, was adopted as the cut-off point for the age category.” (P4 L153-154)
Comment #4
- 4- Table 1 is very confusing, I highly recommend revising it for clarity to the readers.
Response #4
Thank you for pointing this out. For Table 1, we have simplified it by removing the percentage of men in total and the percentage of people aged 60 and over. (Table 1)
Comment #5
- 5- Table 3: I do not understand why risk ratios of metabolic syndrome in the evacuation area was compared with the Aizu area in particular.
Response #5
Thank you for pointing that out. As shown in P4 L163-164, the Aizu area (Mountainous area) is the furthest away from the accidental nuclear power plant in Fukushima Prefecture and was the least affected by the earthquake. For this reason, the evacuation area was compared to the Mountainous area. We have added this information to the analysis section as it was not fully explained. “mountainous area, furthest away from the nuclear power plant in Fukushima and was the least affected by the earthquake,”(P4 L163-164)
Comment #6
- 6- I think the results of this paper are valuable, however, the paper is missing a clear implications and recommendations to reduce cardiovascular risk factors following a disaster.
Response #6
Thank you for pointing that out. We have followed your suggestion and added some implications and recommendations for reducing cardiovascular risk factors after a disaster as follows
“There is a concern that the diseases related to these lifestyle related diseases, such as cardiovascular diseases [26], dementia, and so on will increase in the future, especially among residents of the evacuation area, and it is important to support, particularly with regard to evacuees, men and the elderly and continue to observe and to explore the detailed risk factors in Fukushima residents for the prevention of these diseases. “ (P13 L349-353). 
Reviewer 2 Report
General comments: Generally, this study tended to answer two important questions: 1) the long-term effects of evacuation due to Great East Japan Earthquake on metabolic syndrome; 2) the difference in the effects among different gender and age groups. The research questions are meaningful both theoretically and practically for public health and disaster risk management. Nevertheless, several concerns should be addressed properly before the paper can be published in the International Journal of Environmental Research and Public Health.
First, I suggest the authors to rewrite the abstract with a focus on background, objectives, methodology, main findings and conclusion. Please add a sentence which shows the necessity of the study. Specifically, the abstract of the paper does not point the highlight of the research. According to the title of the article, I guess that the authors' core concern is the impact of evacuation on the long-term trend of metabolic syndrome after the Great East Japan Earthquake. Differences in metabolic syndrome between regions should be highlighted. However, the first sentence of the current abstract juxtaposes four factors: time period, area, gender, and age group. Moreover, the abstract is too descriptive and lengthy. The authors should briefly introduce the main results, and do not need to report the specific values of confidence intervals.
Second, there are insufficient comments on the literatures in the “Introduction” section. What are the methods and findings of existing studies on long-term trends in metabolic syndrome? The authors do not elaborate on them in the introduction. Please provide a concise analysis of existing research, clearly show the knowledge gaps identified and link them to the paper goals.
Third, the authors need to provide the contributions of this study more specific. The last sentence of the “Introduction” section points out that the results of the study will provide important enlightenment for possible disasters in the future. This statement is too general, and does not specifically introduce what the concrete content of the enlightenment is.
Fourth, changes in lifestyle and deterioration of mental health are two possible mediating factors proposed by authors when explaining the mechanism behind the research results. However, these two mediating mechanisms have not been proved by empirical research. Therefore, it is suggested that the authors adjust the expression in the first sentence of the fifth paragraph of "Discussion" or support the above inference with data.
Last but not least, I suggest the authors add line numbers when they re-submit it. It will be easer for reviewers to make comments.
Specific comments:
1. “Figure 1. Mountainous area, central area, coastal area, and evacuation area in Fukushima prefecture.”.
Figure 1 cannot be found in either the manuscript or the supplementary.
2. Line 4 to 6, Page 2: “Studies conducted to date have reported that the incidence of cardiovascular disease and its risk factors has increased after a large-scale disaster in Western countries and Japan [1–7].”
Please eliminate multiple references and characterize each reference individually. It can be done by mentioning 1 or 2 phrases per reference to show how it is different from the others and why it deserves mentioning.
3. “In the United States and Europe, it has been reported that the incidence of cardiovascular disease and its risk factors increased immediately after a large-scale disaster such as an earthquake [1].”
The situation in Europe does not seem to be mentioned in the citation.
4. “Nevertheless, only a few studies have investigated the long-term impact of a large-scale disaster on health and the time it takes for the impact to improve.”
Please provide the citations and describe the findings.

Author Response
Response to Comments
We would like to thank you sincerely for your valuable and useful comments. We have taken your comments into account and have revised the manuscript based on your advice. We hope that you will read our response and recheck the manuscript.
Reviewer #2
Comment#1
First, I suggest the authors to rewrite the abstract with a focus on background, objectives, methodology, main findings and conclusion. Please add a sentence which shows the necessity of the study. Specifically, the abstract of the paper does not point the highlight of the research. According to the title of the article, I guess that the authors' core concern is the impact of evacuation on the long-term trend of metabolic syndrome after the Great East Japan Earthquake. Differences in metabolic syndrome between regions should be highlighted. However, the first sentence of the current abstract juxtaposes four factors: time period, area, gender, and age group. Moreover, the abstract is too descriptive and lengthy. The authors should briefly introduce the main results, and do not need to report the specific values of confidence intervals.
Response #1
Thank you for your important remarks. We have rewritten the abstract according to your suggestions as follows. “Abstract: An increase in lifestyle-related diseases has been reported in Fukushima Prefecture since the Great East Japan Earthquake. However, there are no reports on the long-term trends of lifestyle-related diseases in Fukushima Prefecture as a whole for evacuated areas and other areas. Therefore, the aim of this study was to examine the long-term trends in the prevalence of metabolic syndrome before and after the Great East Japan Earthquake in Fukushima Prefecture according to areas using a national database. The target population was around 330,000-440,000 people each year; Fukushima Prefecture residents aged 40-74 years who underwent specific health check-ups in 2008-2017 participated in the study. Fukushima Prefecture was divided into mountainous, central, coastal and evacuation area, and the prevalence of metabolic syndrome in each fiscal year was calculated by gender and age group for each area and compared before and after the disaster and between areas using a Poisson regression model. Prevalence increased significantly throughout the observation period, particularly in evacuation area. Age- and gender-adjusted prevalence rates significantly increased from 16.2% in 2010 to 19.5% in 2012 (prevalence ratios=1.21) and 20.4% in 2017 in evacuation area. In other areas, Coastal area had the largest increase with 17.9 % (2017), Central area; 16.5 % (2017) and Mountainous area; 18.3 % (2016). The post-disaster increase was particularly high among men and the older age group. The prevalence of metabolic syndrome increased rapidly after the disaster, especially in evacuation area, and continued 6-7 years later. Long-term monitoring and measures to prevent lifestyle-related diseases are needed after major disasters, especially in evacuated areas, among men and the older age group.“ (P1 L22-40)
Comment #2
Second, there are insufficient comments on the literatures in the “Introduction” section. What are the methods and findings of existing studies on long-term trends in metabolic syndrome? The authors do not elaborate on them in the introduction. Please provide a concise analysis of existing research, clearly show the knowledge gaps identified and link them to the paper goals.
Response #2
Thank you for your important remarks. We elaborated on them and showed the gaps in the introduction as you suggested. (p2 L48-94)
Comment #3
- Third, the authors need to provide the contributions of this study more specific. The last sentence of the “Introduction” section points out that the results of the study will provide important enlightenment for possible disasters in the future. This statement is too general, and does not specifically introduce what the concrete content of the enlightenment is.
Response #3
Thank you for your useful suggestions. In accordance with your remarks, we have added a sentence specifically indicate the contribution of this study in terms of the specific content of awareness-raising in the Introduction and Discussion section. Introduction “ If metabolic syndrome, previously identified as a risk factor for lifestyle-related diseases in the immediate aftermath of the disaster, persists over the long term and its impact varies by evacuation status or other characteristics, it will be clear that continuing support for the people of Fukushima and separate support for each background is needed. The findings of this study will provide important insights into the long-term effects of evacuation due to disasters, including earthquakes on metabolic syndromes and possible subsequent cardiovascular diseases, and have important implications for measures against possible future disasters.”(p2 L87-94)
Discussion “The long-term impact of the disaster on metabolic syndrome was greater among men than among women, and in some areas, it was greater among older age group than among younger age group. The above-mentioned effects on work, lifestyle and mental health, as well as attitudes towards health, may differ according to gender and age group, and should be utilized in future disaster prevention measures for lifestyle-related diseases according to gender and age groups. When the results of this study are considered together with the results of previous Fukushima health management survey and other reports, various effects of the disaster are considered to have affected lifestyle-related diseases through changes in lifestyle and deterioration of mental health and these effects have not been improved, or that even if they have been improved, they have not led to improvement of lifestyle-related diseases, which might be the reason that the prevalence of lifestyle-related diseases has not decreased, especially in the evacuation area.
There is a concern that the diseases related to these lifestyle-related diseases, such as cardiovascular diseases [30], dementia, and so on will continue to be higher among people in evacuation areas, and that support for people in evacuation areas, men and the elderly needs to be strengthened. Also, it is important to continuously monitor and explore the detailed risk factors in people in Fukushima for the prevention of these diseases. Moreover, it is also clear that long-term support, especially for people in the evacuation area, men, and the elderly in particular is needed in the event of a possible disaster in the future. “ (P12 L336-P13 L355)
Comment #4
Fourth, changes in lifestyle and deterioration of mental health are two possible mediating factors proposed by authors when explaining the mechanism behind the research results. However, these two mediating mechanisms have not been proved by empirical research. Therefore, it is suggested that the authors adjust the expression in the first sentence of the fifth paragraph of "Discussion" or support the above inference with data.
Response #4
Thank you for your valuable suggestions. In accordance with your remarks, the wording of the first sentence in the fifth paragraph of the 'Discussion' section has been corrected
“When the results of this study are considered together with the results of previous Fukushima health management survey and other reports” (P12 L342-343)
Specific comments #1
“Figure 1. Mountainous area, central area, coastal area, and evacuation area in Fukushima prefecture.”.
Figure 1 cannot be found in either the manuscript or the supplementary.
Response #1
We are very sorry about Figure 1. We will check it carefully when we upload the manuscript.
Specific comments #2
- Line 4 to 6, Page 2: “Studies conducted to date have reported that the incidence of cardiovascular disease and its risk factors has increased after a large-scale disaster in Western countries and Japan [1–7].”
Please eliminate multiple references and characterize each reference individually. It can be done by mentioning 1 or 2 phrases per reference to show how it is different from the others and why it deserves mentioning.
Response #2
Thank you very much for your valid advice. In accordance with your suggestions, we have added a description of the content of each piece of literature and characterized each literature individually. (P2 L 48-94)
Specific comments #3
- “In the United States and Europe, it has been reported that the incidence of cardiovascular disease and its risk factors increased immediately after a large-scale disaster such as an earthquake [1].”
The situation in Europe does not seem to be mentioned in the citation.
Response #3
Thank you for pointing this out. The relevant wording has been corrected. (P2 L50)
Specific comments #4
- “Nevertheless, only a few studies have investigated the long-term impact of a large-scale disaster on health and the time it takes for the impact to improve.”
Please provide the citations and describe the findings.
Response #4
Thank you for pointing this out. We have corrected and added the following sentence for the relevant part in the discussion section as you indicated.
“on the other hand, for long-term health outcome after the disaster, in the systematic review of 58 articles examining post-disaster health hazards, most of which reported increased cardiovascular disease, mortality and its risk factors, such as diabetes and obesity, authors concluded that attention is needed on the detrimental indirect long-term effects on cardiometabolic health [14]. For example, after Katrina struck in 2005, a 12-year follow-up of 21,079 pre-disaster and 84,751 post-disaster subjects reported an increase in acute myocardial infarction, or cardiac death, diabetes and dyslipidemia after the disaster [15], and also emphasize the importance of long-term monitoring of post-disaster health. However there has been no long-term follow-up study of lifestyle-related disease trends after the 2011 Great East Japan Earthquake, tsunami and nuclear accident, comparing evacuated areas with other areas.” (P11 L283- P12 L293)
Reviewer 3 Report
Authors present an interesting research about the long-term trend of health problems in relation to the Fukushima disaster, considering them as an indirect (if not, please clarify) impact of the disastrous event.
The research is inter-multidisciplinary and related also to urban and territorial planning aspects. In particular, do authors explore the environmental and antrhopogenic characteristics of the evacuation areas (natural features, urban functions, air pollution, and so on) as possible trigger factors increasing the metabolic syndrome?
Moreover, I suggest to implement the literature review considering international papers-books-authors not only focusing on Japan-Fukushima case study but on other reality in order to highlight the existing/estimated/possible links among natural disasters - Healthy City concept/paradigm/parameters - lifestyle changes - health problems and deseases.
Minor issues: 1) join conclusion section with discussion (otherwise please expand the conclusions); 2) figure 1 is missing; 3) to ease the reading, I think it could be better to put in tables all the values related to the different survey years (see sections 2.1; 3.1; 3.2).
Author Response
Response to Comments
We would like to thank you sincerely for your valuable and useful comments. We have taken your comments into account and have revised the manuscript based on your advice. We hope that you will read our response and recheck the manuscript.
Reviewer #3
comments #1
The research is inter-multidisciplinary and related also to urban and territorial planning aspects. In particular, do authors explore the environmental and antrhopogenic characteristics of the evacuation areas (natural features, urban functions, air pollution, and so on) as possible trigger factors increasing the metabolic syndrome?
Response #1
Thank you for pointing that out.
We agree with you that environmental and anthropogenic characteristics are one of the important factors triggering the increase in the prevalence of metabolic syndrome as you mentioned. Since main purpose of this study was to clarify the increase in metabolic syndrome after the earthquake, especially in the evacuated areas, we did not examine the factors one by one. However, we have added the content to the discussion section. “In addition to these, various environmental and anthropogenic characteristics associated with evacuation may also be interrelated and influence the increased prevalence of the metabolic syndrome.” (P12 L333-335)
comments #2
Moreover, I suggest to implement the literature review considering international papers-books-authors not only focusing on Japan-Fukushima case study but on other reality in order to highlight the existing/estimated/possible links among natural disasters - Healthy City concept/paradigm/parameters - lifestyle changes - health problems and diseases.
Response #2
 Thank you for your important remarks. We have added content and references for non-Japanese cases to the discussion part.
“Victims of natural disasters such as earthquakes and extreme stress have been reported to trigger acute myocardial infarction and sudden cardiac death in the other countries [18]. After earthquakes in Croatia, increased rates of suicidal thoughts, susceptibility to PTSD and depressive symptomatology have also reported followed by increased numbers of patients with gastric and duodenal ulcers, abdominal pain, bloating and constipation acute myocardial infarction due to possible cardiovascular events caused by acute stress, stroke, arrhythmia, tachycardia Increased incidence of cardiomyopathy and other conditions has been reported, underlining the importance of treating psychological and psychosomatic effects [19].” (P12 L311-320)

Minor comments #1
join conclusion section with discussion (otherwise please expand the conclusions);
Response #1
Thank you for your suggestions. We have revised the information to be in line with your comments. (P13 L368)
Minor comments #2
2) figure 1 is missing;
Response #ï¼’
We are very sorry about this. I will check the Figure 1 is displayed properly when I upload it.
Minor comments #2
3) to ease the reading, I think it could be better to put in tables all the values related to the different survey years (see sections 2.1; 3.1; 3.2).
Response #3
Thank you for your valuable suggestions. We have tabulated all the values and corrected them so that they are in line with your suggestions.
Reviewer 4 Report
please see the attachment

Author Response
Response to Comments
We would like to thank you sincerely for your valuable and useful comments. We have taken your comments into account and have revised the manuscript based on your advice. We hope that you will read our response and recheck the manuscript.
Reviewer #4
comments #1
- Continuous line number is missing. It is difficult to comment without continuous line numbers.
Response #1
We are very sorry for the inconvenience. Line numbers have been added.
comments #2
The title says metabolic syndrome is the effect of evacuation. This should be discussed and justified in the introduction.
Response #2
Thank you very much for pointing this out. In the introduction section, We have added a discussion on the increase in lifestyle-related diseases such as hypertension, diabetes, lipid abnormalities and obesity following major disasters, particularly in evacuated areas, and the possibility that the prolonged evacuation in Fukushima Prefecture may have led to a continued increase in metabolic syndrome rates caused by these diseases. (P2)
comments #3
In line 1-2 of Introduction it is mentioned that along with the earthquake a tsunami and dangerous nuclear accident were also occurred. Thus, effects of earthquake, tsunami and nuclear accident on health hazard (metabolic syndrome) should be discussed.
Response #3
Thank you for your important remarks. We have added a discussion on the impact of earthquakes, tsunamis and nuclear accidents on health hazards in the introduction section (P2) .
comments #4
- Why people evacuated the area? Is it due to the nuclear accident?
Response #4
Thank you for your comment. Yes, the evacuation was originally due to the earthquake, tsunami and nuclear accident, but it was the nuclear accident that subsequently necessitated ongoing evacuation. We have added an explanation in the introduction as it was not clear enough. “Despite more than 10 years have passed since the disaster, more than 35 000 people in Fukushima have been forced to evacuate due to the nuclear accident (as of March 2022)” (P2 L80-82)
comments #5
In line 3 of Introduction, it is mentioned that 164,865 local residents evacuated the area whereas in line 12 of Introduction and in Discussion, >35000 is mentioned.
Response #5
Thank you for pointing this out. As indicated in the text, 164,865 is the maximum number of evacuees in 2012 and over 35 000 is the latest number of evacuees as of 2022.(P2 L48, P2 L80-83)
comments #6
No. of studied people are considered as year-wise. Thus, total number is confusing as many of the people will be common for each year.
Response #6
Thank you for pointing this out. We have removed the information about the cumulative total number to avoid confusion. (P3 L98-103)
comments #7
- The study represents year-wise prevalence of metabolic syndrome. It would be awesome if there is any representation of long term metabolic syndrome statistics for a sample group of people, i.e. the condition of same people from 2008 to 2017.
Response #7
Thank you for your suggestion. In accordance with your suggestion, we calculated the prevalence of metabolic syndrome over time for each area only for those who had been continuously examined from pre-disaster, 2010, to 2017, when the follow-up was completed, and presented the results as Supplemental Figure1. The results showed that the rate of increase from pre-disaster to post-disaster 2011 was still highest in the evacuation area, followed by the Coastal area, Central area, and Mountainous area. The main analysis in this study was conducted using the overall number of people, as it was difficult to get an overall picture when only those who continue to receive specific health check-ups are included in the study, due to the smaller number of subjects.
comments #8
Figure 1 is missing.
Response #8
Thank you for pointing that out. We are very sorry about this. I will check it carefully when I upload it.
comments #9
The criteria 1 (abdominal circumference) for classification (section 2.2.2) should be checked.
Response #9
Thank you for pointing this out. In this study, the Japanese criteria for metabolic syndrome by The Japan Society for the Study of Obesity (JASSO) were adopted. The criteria for abdominal circumference are therefore 1. abdominal circumference: ≥85 cm for men and ≥90 cm for women.
comments #10
What is significance of the (claimed) sensitivity analysis? It is just another results with slight different classification.
Response #10
Thank you for pointing this out. In this study, the Japanese criteria for determining metabolic syndrome were adopted and examined. We considered that there may be cases where the Japanese criteria are difficult to understand, so we checked whether similar results could be obtained in the international criteria of the metabolic syndrome.(supplemental table)
comments #11
Characteristics of the participants (section 3.1) and the Table 1 may be presented in the Materials & Methods section.
Response #11
Thank you for pointing this out. As you pointed out, it is possible that the characteristics of the participants and the Table 1 be shown in the Materials & Methods section, but in this study, authors prefer it is shown in the Results section, and have done so. Thank you for your understanding.
comments #12
In Table 1, what is the necessity to mention nos. and percentage (%) of men? % can be omitted (as women is in nos. also).
Response #12
Thank you for pointing this out. We have removed it as you indicated.
comments #13
Is there any significant difference in Table 2 and Table 3? It is just slight different representation.
Response #13
Thank you for pointing this out. Table 2 is a comparison between before and after the earthquake, while Table 3 is a comparison between the mountainous area where the impact of the earthquake is considered to have been relatively small, and other areas of Fukushima Prefecture including evacuation area, We have prepared both tables as we believe they are necessary.
Round 2
Reviewer 1 Report
The authors have addressed my comments and I have no further comments.
Reviewer 4 Report
I would like to thank the authors for considering my comments. Although I still believe there is room for improvement of the paper in its current form; however, I don't have any further comments for the publication.